# Prevalence, determinants, and antimicrobial susceptibility patterns of Campylobacter infection among under-five children with diarrhea at Governmental Hospitals in Hawassa city, Sidama, Ethiopia. A cross-sectional study

Yeshareg Behailu, Siraj Hussen‡, Tsegaye Alemayehu‡, Mulugeta Mengistu, Demissie Assegu Fenta *

School of Medical Laboratory Science, College of Medicine and Health Science, Hawassa University, Hawassa, Sidama, Ethiopia

☯ These authors contributed equally to this work.
‡ These authors also contributed equally to this work.
* demissieasegu@yahoo.com

## Abstract

### Background

Campylobacteriosis, is a zoonotic bacterial disease observed with a rising worldwide. It is becoming the most commonly recognized cause of bacterial gastroenteritis in under-five mortality in recent years. This study was done to determine the prevalence and determinants of *Campylobacter* infection among under-fives with acute watery diarrhea.

### Methods

This institutional-based cross-sectional study was conducted at governmental and private health institutions in Hawassa city. All outpatient under-five children who met the inclusion criteria from April 2021 to August 2021 were enrolled in this study. Demographic and clinical data were obtained using a standardized data collection tool. Stool samples were collected from each participant with a sterile container and inoculated on a *campylobacter* agar media. The isolates were identified by using biochemical tests and a disc diffusion technique was performed to determine the antimicrobial sensitivity patterns of the isolates. Data were entered and analyzed using SPSS version 21. Descriptive and Logistic regression analysis was applied to determine the determinants *of Campylobacter* infection. P-value < 0.05 was considered statistically significant.

### Results

A total of 235 under-five children were enrolled in this study with a 100% response rate. Of these 130 (55.3%) and 105(44.7%) were males and females respectively with the age range

**Data Availability Statement:** The datasets used and/or analyzed during the current study are available within the paper.

**Funding:** The authors received no specific funding for this work.

**Competing interests:** The authors have declared that no competing interests exist.

**Abbreviations:** AGH, Adare General Hospital; ATCC, American Type Culture Collection; C. *coli*, *Campylobacter coli*; C. jejuni, *Campylobacter jejuni*; CLSI, Clinical Laboratory Standard Institute; HIV, Human Immunodeficiency Virus; HU, Hawassa Universit; MDR, Multi-Drug Resistance; MHA, Muller Hinton Agar; SOPs, Standard Operating Procedures; SPSS, Statistical Package for Social Sciences.

of 2 months to 60 months with the mean age of 25 months. The majority of the 150 (63.2%) were rural residents. Of 235 under-fives with acute watery diarrhea, 16 (6.8%) patients were found to have Campylobacter infection with (95% CI, 3.8–10.2%). Consumption of pasteurized milk (AOR: 0.12; 95% CI 0.02–0.75, P<0.05), presence of domestic animals like cats, hens, and cows (AOR: 0.09: 95% CI 0.01–0.67, P<0.05), absence of handwashing practice before food preparation (AOR: 3.63, 95% CI 1.15–11.46, P<0.05) showed significant association with campylobacter infection.

The antimicrobial susceptibility patterns of the isolated bacteria were 100% sensitivity to Azithromycin, Chloramphenicol, and Gentamicin, however; it was 100% resistant to Cephalothin. The associations of socio-demographic, environmental, and behavioral factors were compared and consumption of unpasteurized milk, the presence of domestic animal like the cat was significantly associated.

## Conclusion

*Campylobacter* infection showed a comparatively low prevalence in under-fives with acute watery diarrhea. In this study contact with cats, consumption of unpasteurized milk were associated with *Campylobacter* infection. The treatment approach of *Campylobacter* infection must consider the sensitivity profile of antibiotics as indicated in the study. We, therefore, recommend further studies to determine the species responsible for *Campylobacter* infection with other co-morbidities and the susceptibility pattern for each species to indicate appropriate antibiotic therapy.

## Introduction

Campylobacter species are small gram-negative, non-spore-forming, helical bacteria with a distinctive 'darting' motility, thermotolerant with a single polar flagellum, and grow in micro-aerobic conditions within the range of 30–42˚C and are catalase and oxidase positive [1]. Campylobacter spp. can be found in the reproductive organs, intestinal tracts, and oral cavity of animals and humans, and the bacterium has increased extremely; and at present, it consists of 17 species and 6 subspecies, many of which might be associated with human disease [2].

*Campylobacter* epidemiology results should be liked with its virulence gene characterization. Although the molecular basis of the pathogenicity of *Campylobacter* has not been fully elucidated, however; several virulence factors have been identified based on in vitro and in vivo studies [3]. For example, colonization of the intestine requires the ability to move into the mucus layer covering the intestinal cells by the polar flagella allowing them to efficiently penetrate the mucus barrier (chemotaxis and flagellar) [4], the genetic virulence factors connected with the invasiveness of campylobacter are placed on plasmid gene that encodes for secretory system protein [5]. Cytolethal distending toxin (CDT) is required for CDT binding to target cells and for the delivery into the cell interior, the thermal stress response of the bacteria by the induction of the expression of heat shock proteins (HSPs) which are important for thermotolerance, putative adhesion factors such as pili and fimbriae characterized by role in adhesion in the intestinal wall [3].

*Campylobacter* infection is frequently asymptomatic, has become one of the most common causative agents of both diarrheal (gastroenteritis) and systemic diseases such as endocarditis, septicemia, cholecystitis, urinary tract infections in humans. It is responsible for 400–500

million cases of infection [6] and greater than 21,000 deaths [7] each year worldwide. Acute infection by Campylobacter can cause serious long-term consequences, such as peripheral neuropathies, Guillain–Barre syndrome [1], Miller Fisher syndrome [8], and functional bowel diseases or irritable bowel syndrome [9]. Diarrhea is highly prevalent in sub-Saharan African countries including Ethiopia [10], resulting from the highest rates of child mortality [11].

Poor hygiene, sanitation, and proximity of man and animals such as including poultry, beef, sheep, goats, cats, and dogs in developing countries facilitate a frequent acquisition of *Campylobacter* infections [12] these factors are responsible for the high prevalence of the disease in these countries Recent studies report a wide range (5–49%) of *Campylobacter* prevalence in healthy sheep and goats. Human exposure can come through direct contact with animal food [13,14]. Flies play a crucial role in the transmission of *Campylobacter* infection from contaminated sources [15]. Other factors have been associated with the occurrence of *Campylobacter* infections, for example, consumption of red meat, grapes, having a chronic illness [16] drinking unpasteurized milk, contaminated water, type of water source, eating prepared salad, latrine usage, bottle feeding, and nutritional status [10]. Consequently, colonization of different reservoirs by *Campylobacter* poses an important risk for humans through the shedding of the pathogen in livestock waste and water sources, environment, and food [17]. Various preventive techniques were reported in the literature including hygiene and sanitation, diet, medications, and supplements classified as health care, breastfeeding, immunization, supplemental zinc, and probiotics [18]. Treatment and prevention of diarrhea can be done at home by primary caregivers, and their role is vital in health promotion, disease prevention, and patient care [19] Prevention practice of caregivers is important and can prevent diarrhea-related child morbidity and mortality.

Multidrug resistance has been increased all over the world that is considered a public health threat. Several recent investigations reported the emergence of multidrug-resistant bacterial pathogens from different origins including humans, birds, cattle, and fish that increase the need for routine application of the antimicrobial susceptibility testing to detect the antibiotic of choice as well as the screening of the emerging MDR strains [20,21]. From a study carried out at the National Referral Laboratory of the Ethiopian Institute of Public Health majority of the isolates showed high levels of antibiotic resistance to commonly prescribed antimicrobials among human urine [22], From a study conducted on the prevalence, multidrug-resistance genes of *E. coli* in Egypt among cattle, 50% of the recovered strains were multidrug-resistant (MDR) for different commonly prescribed drugs [23]. S. aureus strains were isolated from catfish, out of them methicillin resistance S. aureus (MRSA) strains were identified as MRSA in Egypt [24], and a similar study conducted in Egypt from fish on Pseudomonas aeruginosa strains exhibited multi-drug resistance[25] (MDR) From a study conducted at butcher houses in Hawassa city from raw cattle meat, 33.3% of the isolated bacteria were multidrug resistance [26]. Methicillin-resistant *Staphylococcus aureus* (MRSA commonly causes severe infectious diseases, including food poisoning, pyogenic endocarditis, suppurative, pneumonia, otitis media, osteomyelitis, and pyogenic infections of the skin, soft tissues[27]. As indicated in different studies drug resistance conditions in developing countries like Ethiopia might become worse as a result of widespread and unrestricted use of antibiotics as well as inadequate research into antimicrobial resistance. The increasing rate of human infections caused by antimicrobial resistance strains of *Campylobacter* makes clinical management of cases of campylobacteriosis more difficult by prolonging the infection and compromising the treatment [9].

There is evidence that indicates a rise in the global incidence of *Campylobacteriosis* in the past decade [28]. Epidemiological data from Africa, Asia, and the Middle East also showed that *Campylobacter* infection was endemic in these regions. The number of cases of *Campylobacter* has increased in North America, Europe, and Australia [28]. In the United States, it is

estimated to cause 1.3 million human illnesses every year [29]. Studies conducted in Peru 24.9% [30] Pakistan 7% [31] and Israel 53.3% [32] had also demonstrated that *Campylobacter* was the major cause of dysentery among children under the age of 5 years.

In Africa, studies conducted from Nigeria reported the highest prevalence of *Campylobacter* (62.7%) [33] followed by Malawi (21%) [34], two South Africa studies (Vhembe district, (20.3%) [35], and Durban,(21%) [36]), Kenya (16.4%) [37], and Rwanda (15.5%) [38]. In Ethiopia (14.5%) [39,40] of children aged less than five years with diarrheal cases had indicated that campylobacteriosis is an emerging bacterial infection. Furthermore; in Ethiopia, diarrheal disease is a major public health problem, and it is one of the top 15 countries in which nearly three-fourths of child deaths occur due to diarrhea [41]. A few studies reported from Ethiopia also revealed that *Campylobacter* infection was a common cause of childhood diarrhea [42]. Additionally, morbidity reports and community-based studies had revealed that diarrheal diseases are major causes of infant and child mortality and morbidity in the country. From estimated 39,000,000 episodes of diarrhea per year, 230,000 deaths occur in children below five years of age [39]. In our study area, there are indicators of the emergence of *Campylobacter infection* consistently as one of the key contributors to diarrhea and death due to diarrhea [9]. As indicated in the previous studies, the incidence and prevalence of *Campylobacter* infection had increased in both developed and developing countries over the last 10 years, and still, it is one of the most important infectious diseases that is likely to challenge global health in the years to come. The absence of a national surveillance program about the causative organism, its underestimation, lack of a recent report, and well-organized microbiological laboratory make it difficult to give an accurate picture to determine the prevalence, its determinants, and antimicrobial susceptibility patterns of the organism. There are no recent reports on these identified gaps, in the southern region of Ethiopia on Campylobacter infection.

Therefore, this study was aimed to determine the prevalence, its determinants, and antimicrobial resistance of *Campylobacter* infection among under-five children with diarrhea in the study area.

## Materials and methods

### Study area

The study was conducted at Hawassa University Comprehensive Specialized Hospital (HUCSH), Adare General Hospital (AGH), and Bushello health Center in Hawassa city, Sidama regional state located 275 km to the South of Addis Ababa on the shoreline of Lake Hawassa. Based on the information obtained from the webpage of the city administration, the city had a total population of 328, 283 of these 51% and 49% are males and females respectively [43]. Hawassa University comprehensive specialized Hospital and Adare general Hospital are the only governmental Hospitals found in the city and Bushello health center is a non-governmental health center. Hawassa University comprehensive specialized Hospital is serving about 18 million people visiting the hospital from the nearby regions of southern nation's nationalities, Oromiya and Somalia regions. The Hospital is used as a teaching hospital for Medical and Health Sciences students, research and training center. The department of pediatric care unit was among one of the subunits found in the Hospital to provide both inpatient and outpatient services. The other study areas, AGH and Bushello health center are served both inpatient and outpatient pediatric services.

### Study design and period

An Institutional based cross-sectional study was conducted at HUCSH, AGH, and Bushello health center from April 30, 2021, to August 30, 2021.

## Population

**Source population.**  All under-five children and their caregivers who visited the pediatric outpatient department of HUCSH, AGH, and Bushello health center during the study period were the source populations

**Study population.**  All randomly selected under-five children with acute diarrhea and their caregivers who visited the Pediatric outpatient department of HUCSH, AGH, and Bushello health center during the study period and whose guardians gave written consent and assent to participate in the study were included in the study. However; Under-five children with diarrhea and on antibiotic treatment during data collection, treated with antibiotics within the last two weeks, and severely ill were excluded from the study

**Sample size determination.**  The sample size was determined using a single population proportion formula by taking 16.7% from the prevalence study of *Campylobacter* infection conducted in Jimma, Ethiopia [40].

$$n = \frac{Z^2 p(1-p)}{d^2}$$

n = Minimum sample size

Z = Standard deviation (SD) at 95% confidence interval (CI) (1.96)

P = Estimated prevalence of *Campylobacter* species (16.7%)

d = Margin of error (5%)

$n = \frac{(1.96)^2(0.167)(1-0.167)}{(0.05)^2} = 214$. By adding a 10% non-response rate, the final sample size calculated was 235. To maximize the sample size, variables from other related studies [44] were taken into consideration. However, none of the corresponding sample sizes were found to be greater than the calculated sample size for prevalence

## Sampling technique

Hawassa University Comprehensive Specialized Hospital, AGH, and Bushello health center were selected purposely due to their high patient flow rate annually. The annual diarrheic under-five flow of HUCSH, AGH, and Bushello health center was estimated to be 540, 780, and 650 respectively. An equal distribution of cases to 12 months brought the expected monthly average flow to 45, 65, and 54 respectively. Equivalently, the total number of diarrheic under-five children expected to visit HUCSH, AGH during the study period (4 months) was 180, 260, and 216 respectively. Probability proportional to Size (PPS) was applied to allocate the corresponding study subjects from each study area. Finally, a systematic random sampling technique was applied to select the study participants. The sampling interval was calculated by dividing the entire expected diarrheic under-five children by the desired sample size. A simple random sampling technique was used to select the first study subject.

## Operational definition

**Diarrhea.**  is the passage of three or more loose or liquid stools per day, or more frequently than is normal for the individual, or within two weeks before the survey as reported by the mother/ caregivers of the child [45].

**Data collection procedure.**  Data on socio-demographic, environmental, behavioral, and clinical variables were collected by face-to-face interviews from guardians/caretakers of the study participants using a structured questionnaire adapted from the Ethiopian demographic health survey (EDHS) and previous studies. Trained BSc Nurse to conduct a face-to-face

interview and medical laboratory technologist who collect, handle, and transport stool samples for the microbiological study were trained for one day.

**Specimen collection and processing.** *Bacterial isolation and identification*. Approximately 2 grams of fresh stool specimen was collected from each child using a leakproof and clean stool screw cup with a sterile applicator stick. Each sample was labeled with a unique code taken from the questionnaire and transported immediately to Hawassa University, College of Medicine and Health Sciences Medical Microbiology Laboratory using Cary Blair transport media. The specimens were inoculated on CHROMagar$^{TM}$ (*Campylobacter* Agar Base $_+$ supplemented with sodium pyruvate, cefoperazone, vancomycin, cycloheximide, and blood free charcoal-based selective media called Sodium Deoxycholate agar (CCDA) (BioMerieux, Paris, France), and Blood contained media selective for Campylobacter (BioMerieux, Paris, France) [46, 47].

This is a medium in which blood was replaced by charcoal, ferrous sulphate, and sodium pyruvate. The peptic digest of animal tissue, beef extract, and casein enzymic hydrolysate provides organic nitrogen to the organisms. Sodium chloride maintains the osmotic balance. Sodium Deoxycholate inhibits the growth of most gram-positive microorganisms. The selectivity of the media was achieved by using cefoperazone and then through incubation at 42 $^0$ C [46]. The inoculated media were kept in a 2.5-liter anaerobic jar with a Campy-Gen gas generating kit which has microaerophilic atmospheric conditions equivalent to 5% $O_2$ and 10% $CO_2$, 85% $N_2$) (Oxoid CN0025A), which was inserted to maintain the microaerophilic conditions. The jars were incubated at 42˚C and the bacterial growth was examined after 24hrs. then at 48hrs finally at 72hrs of incubation [47].

The colonies with typical morphology were further identified using gram stain, catalase test, oxidase test, susceptibility to Nalidixic acid (30μg) (BioMerieux, Paris, France) and cephalothin (80μg) (BioMerieux, Paris, France) and Sodium hippurate hydrolysis [46,47].

## Antimicrobial susceptibility test

Antimicrobial susceptibility test for Campylobacter was performed using the standard agar disc diffusion method as recommended by Clinical Laboratory Standards Institutions (CLSI2020) [48] The commonly prescribed antimicrobials were obtained from Oxoid at the concentration of amoxicillin with clavulanic acid (30μg), gentamicin (10μg), chloramphenicol (30μg), ciprofloxacin (5μg), norfloxacin (5μg), ceftriaxone (5μg) erythromycin (15μg), clindamycin (15μg), nalidixic acid (30μg), azithromycin, (15μg), cefalotin (30μg) and trimethoprim-sulphamethoxazole- STX (25μg) and incubated at 42˚C for 48 hours in an anaerobic jar using $CO_2$ generating kits.

In brief, 3–4 morphologically identical colonies of bacteria from culture were picked and suspended in sterile normal saline. Turbidity of the broth culture was compared with that of 0.5 McFarland turbidity standards. A loop full of the bacterial suspension was placed at the center of Muller Hinton agar media (Oxoid, LTD) supplemented with 5% sheep blood and evenly spread using a sterile cotton-tipped applicator. After drying, antibiotic discs were placed and incubated at 42˚C for 48 hours in an anaerobic jar using CO2 generating kits. Finally, the diameter of growth inhibition around the discs was measured and interpreted as sensitive (S), and resistant (R) as per the guidelines of the manufacturer. Control strains of E. coli (ATCC 25922) sensitive to all antibiotic being tested was inoculated to evaluate the performance of culture media and antibiotic discs [48].

**Data processing and analysis.** Collected data were coded, cleaned, and entered into SPSS version 21 software packages (IBM® SPSS® Statistics is a powerful statistical software platform. It offers a user-friendly interface and a robust set of features that lets us quickly extract

actionable insights from our data). Descriptive statistic was used to describe the findings of the study. Mean and standard deviation (SD) were used to describe continuous data while frequency and percentage were used to describe categorical variables. Logistic regression was used to examine the association between dependent and independent variables. The association between each independent variable and the dependent variable was first analyzed using binary logistic regression. Those variables with a P-value < 0.25 were entered into multiple logistic regressions. Variables with a P value< 0.05 in multiple logistic regression were considered as significantly associated with the outcome variable.

**Data quality control.**   The quality of data was ensured by using a structured and pretested questionnaire, training was given for data collectors, and strict follow-up was made by the principal investigator. In the meantime, standard operating procedures (SOPs) of the microbiology laboratory were strictly followed. The prepared media was sterilized before inoculation and then checked for its sterility by incubating 5% of the total media at 35–37˚C overnight and observing for any bacterial growth. Those media that showed growth were discarded and prepared again following the correct Standard operating procedures (SOPs). Reference strains such as *Campylobacter coli* (ATCC-33559) and *E. coli* (ATCC-25922) were used to check the performance of the culture media.

## Ethical consideration

Ethical clearance was obtained from the Institutional Review Board (IRB) of Hawassa University. College of Medicine and Health Sciences approved by the ethics committee under the chairperson of Dawit Jember (Assistant professor) with Ref.No.IRB/088/13 (Tell: +-0468209290). Permission letter was also obtained from Hawassa University comprehensive specialized Hospital Chief clinical and academic director office, Adare General Hospital, and Bushello health center Administration. All selected study participants signed the informed consent and assent form. Study participants were also informed about the purpose of the study, the importance of their participation, and withdrawal at any time. Privacy and confidentiality of information given by each respondent were kept and names were not recorded. Clinicians were notified about the results of the tests, and participants with positive test results were treated by the clinicians of the Hospitals. Besides guardians of the study participants were informed about prevention and control measures of campylobacter infection.

## Result

### Socio-demographic and economic characteristics of the participants

A total of 235 under-five children with diarrhea were included in this study with a 100% response rate. Of these 130 (55.3%) were males and 105(44.7%) were females with the age range of two months to 60 months with the mean age of 25 months. Thirty-nine (16.6%) children were younger than one year. The majority 150 (63.8%) of them were rural residents, while 85(36.2%) were urban dwellers. Most of the caregivers were married 218 (92.8%) and had not had any formal education 122 (51.9%). About 182 (77.4%) of caregivers were unemployed, and 125 (53.2%) of caregivers had a family size of < 5 persons. Among the caregivers, 182 (77.4%) of them had a monthly income of less than 2500 ETB (**Table 1**).

### Prevalence and phenotypic characteristics of the recovered isolates of Campylobacter

The techniques employed for differentiating campylobacter isolates phenotypically in this study are based on the presence or absence of biological or metabolic activities expressed by

**Table 1. Socio-demographic and economic characteristics of the participants at governmental hospitals in Hawassa city, Sidama, Ethiopia, 2021.**

| Sociodemographic variables | | Frequency (%) | Positive (%) |
|---|---|---|---|
| Age (in years) | <1 | 39 (16.6) | 2 (0.9) |
| | 1–5 | 196 (83.4) | 14 (5.9) |
| Sex | Male | 130 (55.3) | 10 (4.3) |
| | Female | 105 (44.7) | 6 (2.5) |
| Place of residence | Urban | 85 (36.2) | 6 (2.6) |
| | Rural | 150 (63.8) | 10 (4.3) |
| Educational Status of caregiver | | | |
| | Informal education | 122 (51.9) | 10 (4.2) |
| | Formal education | 113 (48.1) | 6 (2.6) |
| Occupational Status of caregiver | | | |
| | Employed | 53 (22.6) | 5 (2.1) |
| | Unemployed | 182 (77.4) | 11 (4.7) |
| Marital status of the caregiver | | | |
| | Married | 218 (92.8) | 14 (5.9) |
| | Divorced / Widowed | 17 (7.2) | 2 (0.9) |
| Family size | <5 | 125 (53.2) | 9 (3.8) |
| | ≥5 | 110 (46.8) | 7 (3) |
| Monthly income | <2500 | 182 (77.4) | 14 (5.9) |
| | ≥2500 | 53 (22.6) | 2 (0.9) |

the organism. The most popularly used phenotypic methods to differentiate *Campylobacter* isolates include biotyping, serotyping, and multilocus enzyme electrophoresis. In this study, we used the Biotyping method for the identification of bacterial isolates through the expression of metabolic activities. These metabolic activities can include colonial morphology, environmental tolerances, and biochemical reactions. Identification of *Campylobacter* isolates begins with culturing of the bacteria on selective media to observe their colonial morphology. These modified agars use different combinations of antibiotics to discourage the proliferation of any non-*Campylobacter* species or contaminants; however, the selectivity of these media may inhibit the growth of some of the less common species of *Campylobacter*. The thermotolerant nature of *Campylobacter* isolates also discourages the growth of other bacteria on media. Most bacteria, including non-thermophilic *Campylobacter* spp., do not grow well at 42˚C. After growth on a medium in a microaerophilic environment, a suspect colony can then undergo a series of biochemical reactions to further type the strain. Gram staining, oxidase tests, and catalase tests are the easiest techniques to quickly identify *Campylobacter* cells.

Among 235 under-five children whose stool specimens were cultured, *Campylobacter* infection was isolated from 16 (6.8%) with (95% CI, 3.8–10.2%). Ten (4.3%) of them were males and 6 (2.5%) of them were female under-five children were positive for *Campylobacter* infection. Campylobacter infection was higher 14 (5.9%) among children aged between 1–5 years.

The prevalence of Campylobacter infection among children whose families had no formal education was 10 (4.2%). According to the occupational status, the prevalence of 11(4.7%) of *Campylobacter* infection was higher among children whose families were unemployed.

## Environmental health conditions and behavioral practices

The majority 211 (89.9%) of participants responded that they were washing their hands before food preparation, washing utensils before food preparation 204(86.8%), washing hands after cleaning the child 210 (89.4%), having contact with domestic animals 45(19.2%) while they

handle and feed their child and 180(76.7%) of them practiced washing with soap and water before preparing food for the child. The percentage of guardians/caretakers who had a habit of washing hands after assisting/cleaning the child was highest at 210 (89.4%) compared to their counterparts.

Among the study participants, 188 (80%) and 47 (20%) were using pipe and spring water as their source of drinking respectively and none of the study them were using water from unprotected sources. The prevalence of *Campylobacter* infection among study participants who used pipe and spring water as a source of drinking was 12(5.1%) and 45(19.2%) respectively. Almost all respondents 225(95.7%) reported that they had at least a sort of traditional pit latrine, and 63(26.8%) respondents also stated that they had at least one type of domestic animal like dog, cat, sheep, chicken, and cattle **(Table 2)**.

## Clinical factors associated with Campylobacter infection

The most common form of diarrhea noted in this study was watery diarrhea 80 (34%) followed by semi-fluid 79 (33.6%), mucoid 67 (28.6%), and bloody 9 (3.8%). The prevalence of campylobacter infection was higher among children with bloody diarrhea 8 (3.4%) followed by mucoid 6 (2.5%). The majority 167 (71%) of study participants had a frequency greater than 2 times within 24 hours. Among them, 15(6.4%) were found to be positive for campylobacter infection. Other clinical factors observed in the current study were vomiting, fever and abdominal pain were 119 (50.6%), 63 (26.8%), and 53 (22.6%) respectively **(Table 3)**.

## Factors associated with Campylobacter infection

To identify possible factors associated with *Campylobacter* infection, socio-demographic, environmental, and behavioral factors were analyzed using bivariate and multivariate logistic

**Table 2. Environmental health conditions and behavioral practices for *Campylobacter* infection among under-five children with diarrhea at governmental hospitals in Hawassa city, Sidama, Ethiopia, 2021.**

| Behavioral and environmental factors | | Frequency (%) | Positive (%) |
|---|---|---|---|
| Type of water source | Pipe | 188 (80) | 12 (5.1) |
| | Spring | 47 (20) | 4 (1.7) |
| Consume fruit and vegetable | Yes | 133 (56.6) | 15 (6.4) |
| | No | 102 (43.4) | 1 (0.4) |
| Consume only pasteurized milk | Yes | 68 (28.9) | 12 (5.1) |
| | No | 167 (71.1) | 4 (1.7) |
| Have latrine | Yes | 225 (95.7) | 14 (5.9) |
| | No | 10 (4.3) | 2 (0.9) |
| Have domestic animals | Yes | 63 (26.8) | 14 (5.9%) |
| | No | 172 (73.2) | 2 (0.9%) |
| Dog present | Yes | 16 (6.8) | 6 (2.5) |
| | No | 219 (93.2) | 10 (4.3) |
| Cat present | Yes | 21 (8.9) | 9 (3.8) |
| | No | 214 (91.1) | 7 (3) |
| Sheep present | Yes | 10 (4.3) | 2 (0.9) |
| | No | 225 (95.7) | 14 (5.9) |
| Cattle present | Yes | 16 (6.8) | 1 (0.4) |
| | No | 219 (93.2) | 15 (6.4) |
| Chickens present | Yes | 20 (8.5) | 0 |
| | No | 215 (91.5) | 16(6.8) |

**Table 3. Clinical characteristics and distribution of Campylobacter infection among under-five children at governmental hospitals in Hawassa city, Sidama, Ethiopia, 2021.**

| Clinical history | | Frequency n (%) | Prevalence n (%) |
|---|---|---|---|
| Vomiting | Yes | 119 (50.6) | 7 (3) |
| | No | 116 (49.4) | 9 (3.8) |
| Fever | Yes | 63 (26.8) | 5 (2.1) |
| | No | 172 (73.2) | 11(4.7) |
| Abdominal pain | Yes | 53 (22.6) | 4 (1.7) |
| | No | 182 (77.4) | 12 (5.1) |
| Nausea | Yes | 17 (7.2) | 2 (0.9) |
| | No | 218 (92.8) | 14 (5.9) |
| Chills | Yes | 16 (6.8) | 1(0.4) |
| | No | 219 (93.2) | 15 (6.4) |
| Consistency of diarrhea | Bloody | 9 (3.8) | 8 (3.4) |
| | Watery | 80 (34) | 2 (0.9) |
| | Mucoid | 67 (28.6) | 6 (2.5) |
| | Semifluid | 79 (33.6) | 0 |
| Frequency of diarrhea in 24hrs | ≤ Two time | 98 (41.7) | 1(0.4) |
| | > Two times | 137 (58.3) | 15 (6.4) |

regressions were performed. From all possible factors which were considered; consumption of pasteurized milk (AOR: 0.12; 95% CI: 0.02–0.75), presence of domestic animals like cats, hens, and cows (AOR: 0.09: 95% CI: 0.01–0.67), absence of handwashing practice before food preparation (AOR: 3.63, 95%CI: 1.15–11.46, showed significant association with campylobacter infection.

The likelihood of having Campylobacter infection among those who consumed pasteurized milk was less risky than their counterparts. Similarly, those families who had not had a domestic cat were also less likely to have campylobacter infection than those families who had domestic cats. In the current study, no other statistically significant associations were observed with other independent variables (**Table 4**).

## Antimicrobial susceptibility testing

All 16 specimens found to be positive for Campylobacter were tested for antimicrobial susceptibility test for commonly prescribed drugs. Based on these the identified isolates showed that as such they were sensitive to Chloramphenicol, Gentamicin, and Azithromycin. But; Cephalothin (30 µg) was 100% resistant to the isolated bacteria in this study. A higher rate of resistance was also observed with Amoxicillin, clavulanic acid (68.8%), followed by Tetracycline (31.3%), Trimethoprim-sulfamethoxazole (31.2%), Ciprofloxacin, Nalidixic acid (18.8%), and Clindamycin (6.3%) (**Table 5**).

## Multidrug resistance in Campylobacter isolates from patients

Multidrug resistance (MDR) in this study was defined as resistance to three or more classes of antibiotics. 7 isolated species out of (16) Campylobacter isolates were multidrug-resistant (43.7%). Based on their specific antibiotic eleven campylobacter s species were resistant to Amoxicillin with clavulanic acid, five species for Tetracycline, and 4 species for Trimethoprim-sulfamethoxazole as indicated in **Table 6**.

**Table 4. Logistic regression analyses for identifying factors association Campylobacter infection among under-five diarrheic children in Hawassa city, Sidama, Ethiopia, 2021.**

| Variables | Category | No (%) tested | Positive No (%) | COR(95%CI) | P-value | AOR(95%CI) | P-value |
|---|---|---|---|---|---|---|---|
| Age (year) | <1 | 39(16.6) | 2(0.9) | 1 | | | |
| | 1–5 | 196(83.4) | 14(5.9) | 0.70 (0.153–3.223) | 0.650 | | |
| Sex | Male | 130(55.3) | 10(4.3) | 0.73 (0.255–2.071) | 0.551 | | |
| | Female | 105(44.7) | 6(2.5) | 1 | | | |
| Place of Residence | Urban | 85(36.2) | 6(2.6) | 1 | | | |
| | Rural | 150(63.8) | 10(4.3) | 1.06 (0.372–3.036) | 0.909 | | |
| Educational Status | Informal | 122(51.9) | 10(4.3) | 0.63 (0.221–1.788) | 0.384 | | |
| | Formal | 113(48.1) | 6(2.5) | 1 | | | |
| Occupational Status | Employed | 53(22.6) | 5(2.1) | 1 | | | |
| | Un Employed | 182(77.4) | 11(4.7) | 1.62 (0.537–4.887) | 0.392 | | |
| Marital status | Married | 218(92.8) | 14(5.9) | 1.94 (0.404–9.354) | 0.408 | | |
| | Divorced/widowed | 17(7.2) | 2(0.9) | 1 | | | |
| Family size | < 5 | 182(77.4) | 14(5.9) | 0.88 (0.315–2.436) | 0.800 | | |
| | ≥ 5 | 53(22.6) | 2(0.9) | 1 | | | |
| Monthly income | <2500 | 125(53.2) | 9(3.8) | 0.47 (0.104–2.140) | 0.329 | | |
| | ≥2500 | 110(46.8) | 7(3) | 1 | | | |
| Wash hands before preparing food | Yes | 211(89.9) | 12 (5.1) | 3.32(0.97–11.25) | 0.054 | 2.5 (0.08–83.83) | 0.603 |
| | No | 24 (10.2) | 4 (1.7) | 1 | | | |
| Wash utensils before preparing food | Yes | 204 (86.8) | 12 (5.1) | 2.37 (0.71–7.87) | 0.159 | 1.77 (0.39–7.94) | 0.455 |
| | No | 31(13.2) | 4 (1.3) | 1 | | | |
| Consume fruit and vegetable | Yes | 133 (56.6) | 15 (6.4) | 0.08 (0.01–0.60) | 0.014 | 0.26 (0.02–3.01) | 0.284 |
| | No | 102 (43.4) | 1 (0.4) | 1 | | | |
| Consume pasteurized milk | Yes | 68(28.9) | 12 (5.1) | 0.12 (0.03–0.37) | <0.001 | 0.12 (0.02–0.75) | 0.024 |
| | No | 167 (71.1) | 4 (1.7) | 1 | | | |
| Washing hands after assisting the child | Yes | 210 (89.4) | 12 (5.1) | 3.14(0.93–10.62) | 0.065 | 0.76 (0.02–37.4) | 0.889 |
| | No | | | 1 | | | |
| Have latrine | Yes | 225 (95.7) | 14 (5.9) | 3.77(0.73–19.44) | 0.113 | 11.5 (0.67–7.26) | 0.092 |
| | No | 10 (4.3) | 2 (0.9) | 1 | | | |
| Have domestic animal | Yes | 63 (26.8) | 14 (5.9) | 0.04 (0.01–0.18) | <0.001 | 0.37 (0.03–5.07) | 0.454 |
| | No | 172 (73.2) | 2 (0.9) | 1 | | | |
| Dog present | Yes | 16 (6.8) | 6 (2.5) | 1 | | | |
| | No | 219 (93.2) | 10 (4.3) | 12.5(3.79–41.41) | <0.001 | 0.61 (0.09–4.09) | 0.612 |
| Cat present | Yes | 21(8.9) | 9 (3.8) | 1 | | | |
| | No | 214 (91.1) | 7 (3) | 0.05(0.01–0.14) | <0.001 | 0.09 (0.01–0.67) | 0.019 |
| Contact with domestic animals | Yes | 45 (19.1) | 13 (5.5) | 0.04 (0.01–0.14) | <0.001 | 0.33 (0.02–4.51) | 0.402 |
| | No | 190 (80.9) | 3 (1.3) | 1 | | | |

**AOR = Adjusted Odds Ratio, COR = Crude Odds Ratio, No = Number.**

The frequency of multidrug-resistant strains (resistant to two or more drugs) (68.2%) in this study was higher than the previous finding in Ethiopia (20%) [15].

## Discussion

This study was conducted to address three key objectives; determined the prevalence, antibiotic resistance pattern of isolates, and identified factors associated with *campylobacter* infection.

**Table 5. Antimicrobial susceptibility pattern of Campylobacter among under-five children at Hawassa.**

| Classes of Antibiotics | Antimicrobial agents | Susceptibility pattern of Campylobacter isolates | | |
|---|---|---|---|---|
| | | S (%) | I (%) | R (%) |
| Macrolides | Azithromycin (15 μg) | 16 (100) | 0 | 0 |
| | Erythromycin (15 μg) | 8 (50) | 8 (50) | 0 |
| Anti-50S ribosomal | Chloramphenicol (30 μg) | 16 (100) | 0 | 0 |
| Tetracyclines | Tetracycline (30 μg) | 7 (43.7) | 4 (25) | 5 (31.3) |
| Aminoglycosides | Gentamicin (10 μg) | 16 (100) | 0 | 0 |
| Beta-lactamase inhibitors | Amoxicillin with clavulanic acid (10/20 μg) | 0 | 5 (31.3) | 11 (68.7) |
| Folic acid synthesis inhibitors | Trimethoprim-sulfamethoxazole (25 μg) | 9 (56.3) | 2 (12.5) | 5 (31.2) |
| Fluoroquinolones | Ciprofloxacin (5 μg) | 12 (75) | 1 (6.3) | 3 (18.7) |
| Lincosamide | Clindamycin (2 μg) | 10 (62.5) | 5 (31.2) | 1 (6.3) |
| Cephalosporins 1st generation | Cephalothin (30 μg) | 0 | 0 | 16 (100) |
| DNA synthesis inhibitors | Nalidixic acid (30 μg) | 5 (31.3) | 8 (50) | 3 (18.7) |

S: Sensitive; I: Intermediate, R: Resistance, μg: Microgram.

## General patient characteristics

This institutional-based study enrolled 235 patients below the age of 60 months with acute watery diarrhea with the mean age of 25 months, compared to our finding a larger mean age (37.8) than our study was reported from Jimma, Ethiopia [40]. A consistent finding was obtained from a study in Gondar, Ethiopia [39], and Kampala, Uganda [49]. In our study, 74% of the recruited children were below 2 years of age which was a similar finding with studies reported from Baheredar, Ethiopia [50], and Gondar, Ethiopia [39].

## The prevalence of Campylobacter infection

In our study the overall prevalence of *campylobacter* infection among children with acute diarrhea was calculated to be 6.8% with (95% CI: 3.8–10.2%). This finding was lower than the other studies from Jimma Ethiopia 16.7% [40], Gondar Ethiopia 15.4% [39], 11.9%, 14%, and

**Table 6. Multiple drug resistance patterns of Campylobacter isolates in under five diarrhoeic children at Governmental Hospitals in Hawassa city, Sidama, Ethiopia.**

| Classes of Antibiotics | Lists of Antibiotics | No of Campylobacter spp isolated (N = 16) | R0 | R1 | R2 | R3 | R≥4 |
|---|---|---|---|---|---|---|---|
| Macrolides | Azithromycin (15 μg) | | 16 | 0 | 0 | 0 | 0 |
| | Erythromycin (15 μg) | | 16 | 0 | 0 | 0 | 0 |
| Anti-50S ribosomal | Chloramphenicol (30 μg) | | 15 | 1 | 0 | 0 | 0 |
| Tetracyclines | Tetracycline (30 μg) | | 11 | 0 | 0 | 0 | 5 |
| Aminoglycosides | Gentamicin (10 μg) | | 16 | 0 | 0 | 0 | 0 |
| Beta-lactamase inhibitors | Amoxicillin with clavulanic acid (10/20 μg) | | 5 | 0 | 0 | 0 | 11 |
| Folic acid synthesis inhibitors | Trimethoprim-sulfamethoxazole (25 μg) | | 12 | 0 | 0 | 0 | 4 |
| Fluoroquinolones | Ciprofloxacin (5 μg) | | 15 | 1 | 0 | 0 | 0 |
| Lincosamide | Clindamycin (2 μg) | | 15 | 1 | 0 | 0 | 0 |
| Cephalosporins 1st generation | Cephalothin (30 μg) | | 13 | 0 | 0 | 3 | 0 |
| DNA synthesis inhibitors | Nalidixic acid (30 μg) | | 13 | 0 | 0 | 3 | 0 |

R0 = No drug resistance, R1 = Resistant to one drug, R2 = Resistant to two drugs, R3 = Resistant to three drugs, R4 = Resistant to four drugs.

12.9%, in different Kenyan studies [51], Hawassa, Ethiopia 12.7% [52], Malawi 21% [34], South Africa 21% [44], Peru 24.9% [30] and Israel 53.3% [32]. But, it was higher than the other studies conducted in other African studies in Sudan 2% [53], Burkina Faso 1.1% [54], and Zambia 3.5% [55], and Asian studies conducted from India, 3.8% [41], in two other Iranian studies 4.1% [42], and 4.5% [43]. In line with our finding was also reported from studies conducted in Pakistan, 7% [31], Uganda, 9.3% [49], Tanzania, 9.7% [56], and Madagascar, 9.7% [57].

The discrepancy might be related to the socio-economic status, degree of contact with animals and animal products, environmental sanitation, personal hygiene, study period/season, food handling practices, sample collection, transportation, culture conditions, and the laboratory methods used. The contribution of COVID-19 preventive measures such as limited movement/traveling of people from place to place and use of hand sanitizer and frequent hand washing would decrease the prevalence of the infection in our study.

In our study, the distribution of Campylobacter infection between males and females was not statistically significant, which was a similar finding to the study conducted in Ethiopia [58]. Higher rates of infection were detected in rural residents 10 (4.3%) than urban 6 (2.6%) children, which is in line with the findings reported in Yemen [59]. However; the finding was not statistically significant. This may be due to unprotected water sources and the presence of domestic animals in almost all rural households.

The results of this study showed that *Campylobacter infection* was higher among children whose caregivers had no formal education. This is in line with a study conducted in England [60], but a study conducted in New Zealand [61] showed high educational attainment, and homeownership greater than 50% was associated with an increased incidence of *Campylobacter infection*.

Higher rates of Campylobacter infection were observed among children whose caretaker had a habit of washing hands before feeding and preparing foods, cleaning utensils with soap, washing the child with soap and water after defecation. However; no statistically significant association was found with *Campylobacter* infection. This finding was in line with the study conducted in Gondar [39]. In this study, high infection rates were also seen in under-five children whose families had latrines than those who had no latrine in their homes. The contracting finding to our result was reported from Gondar[39] where usage of latrine had statically significant with *Campylobacter* infection

These findings may look contradictory as *Campylobacter infection* can be best prevented by proper food handling, cooking, good hygiene practices, and safe water supplies [62] but sends an alarming message to health and hygiene service providers that hygiene and handling practices being practiced by communities were not adequate.

In our study children who used pipe water as a source had high rates of infection than those who used spring water. But other research findings showed that contaminated water was the most common source of *campylobacter infection* [16]. This difference might be due to lack of personal hygiene, cleaning habit of the water storage materials, using of improperly treated pipe water, due to lack of water source maintenance and cleaning regularly to avoid the buildup of debris and sludge that can cause health problems. Even in developed countries where safe water supply is assured, waterborne outbreaks caused by different species of bacteria including *campylobacter* are common [28]. Likewise, the findings of this study revealed that a high prevalence of *campylobacter* infection was observed among families who got water from protected water sources (pipe and spring) that could be most likely due to cross-contamination and/or poor handling practices. Pipe water or source protection by itself is not a guarantee for water safety unless the water is handled in a way that will keep it free from contamination from the source to consumption.

Smaller infection rates were seen in children who have close contact with cats, which indicates the direct association between Campylobacter infection and cats, as it is already pointed out that direct contact with such animals is a frequent mode of transmission to humans. Consistent; findings to this study were also reported from Gondar [44], Jimma [40], and Burkina Faso [63]. Children who consume pasteurized milk were more vulnerable to campylobacter infection than those who drink unpasteurized milk. This finding showed a statistically significant association with campylobacter infection and similar findings were conducted in Gondar [44], Jimma [40], and Burkina Faso [63].

Different clinical symptoms such as fever and vomiting were not statistically significant in our study. This was also evidenced by studies from Gondar [44] and Jimma

From eleven antibiotics used for susceptibility test against isolates of *Campylobacter* infection in the current study, higher sensitivity (100%) rate was observed for Azithromycin, chloramphenicol, and gentamicin whereas ciprofloxacin was found to be 75% sensitive. An approximately similar finding was observed in a study conducted in Gondar teaching Hospital [39] indicated that chloramphenicol, gentamicin, and ciprofloxacin were 88.6%, 81.8%, and 84% sensitive respectively. On the other hand, this study also revealed that Erythromycin was 50% sensitive. An inconsistent finding was reported from Gondar[39] where erythromycin was resistant to Campylobacter infection (22.7%), this is alarming because erythromycin is the drug of choice for Campylobacter species. Erythromycin resistance was higher than the previous study in Jimma (10%) [58]. and Addis Ababa (2%) [64]. This might be because erythromycin may act as selective pressure and favor the proliferation of resistant strains.

Campylobacter isolates were found to be completely resistant to Cephalothin (100%). Besides, a moderate to low level of resistance was found for trimethoprim-sulfamethoxazole (31.2%), Ciprofloxacin (18.7%), Nalidixic acid (18.8%), and clindamycin (6.3%). A similar pattern of resistances for Cephalothin (100%). was reported from Jimma [58], a higher resistance pattern was observed to trimethoprim-sulfamethoxazole (61.5%) from a study conducted in Ghana [20] (68.4%) from Jimma [58].

Generally, the variation in the resistance level of these antibiotics might be mainly due to differences in strain and culture media. The indiscriminate use of antibiotics in the community, availability/accessibility of antibiotics in almost all locally available pharmacies, and inappropriate use of prescribed antibiotics might have contributed to such higher resistance rates [65]. Lastly speciation, genotype studies were not done so it is difficult for this study to recommend on treatment policy for each species of the bacteria. Despite these limitations objectives of the study were achieved and discussed.

## Conclusion

Campylobacter infection shows a comparatively low prevalence (6.8%) in under-fives with acute diarrhea in the study area compared to other similar studies conducted in Ethiopia previously. This study also showed that contact with domestic animals, consumption of unpasteurized milk were significantly associated with Campylobacter infection which probably transferred resistant strains of the organisms to children through direct contact or environmental contamination. These isolates showed higher resistance to commonly prescribed drugs in the study area. Continuous assessment of the prevalence and antimicrobial susceptibility patterns of *Campylobacter* infection in hospitals and the community is required. Hence, the data may be used as a baseline for large-scale and public health research. The health system actors should show in tandem with veterinary health officials through one health approach to break the spill-over transmission pathogens from animal to human and vice versa. Further large scale studies are required for serotyping and molecular studies should be conducted to

identify the dominant strains responsible for infection and their antibiotic susceptibility pattern to provide evidence, emphasizing the need for strengthening both national and regional multisectoral antimicrobial resistance to control both the campylobacteriosis burden and increase of antimicrobial resistance in the study area and nationwide.

## Acknowledgments

The authors would like to acknowledge Hawassa University, College of Medicine and Health Science, School of Medical Laboratory Science for allowing me to conduct this research. Hawassa University Comprehensive Specialized Hospital, Medical Microbiology Laboratory, Adare General Hospital Laboratory staffs, and Bushello health Center and laboratory staffs for permitting to conduct this research. Finally, our thanks go to the study participants' children and Caregivers/guardians for their willingness to give the necessary information.

## Author Contributions

**Conceptualization:** Yeshareg Behailu, Siraj Hussen, Tsegaye Alemayehu, Mulugeta Mengistu, Demissie Assegu Fenta.

**Data curation:** Yeshareg Behailu, Siraj Hussen, Tsegaye Alemayehu, Mulugeta Mengistu, Demissie Assegu Fenta.

**Formal analysis:** Yeshareg Behailu, Mulugeta Mengistu, Demissie Assegu Fenta.

**Investigation:** Yeshareg Behailu, Siraj Hussen, Tsegaye Alemayehu, Demissie Assegu Fenta.

**Methodology:** Yeshareg Behailu, Siraj Hussen, Tsegaye Alemayehu, Demissie Assegu Fenta.

**Project administration:** Yeshareg Behailu.

**Resources:** Yeshareg Behailu.

**Software:** Yeshareg Behailu, Siraj Hussen, Mulugeta Mengistu, Demissie Assegu Fenta.

**Supervision:** Siraj Hussen, Tsegaye Alemayehu, Mulugeta Mengistu.

**Validation:** Siraj Hussen, Tsegaye Alemayehu.

**Visualization:** Yeshareg Behailu, Tsegaye Alemayehu, Mulugeta Mengistu.

**Writing – original draft:** Yeshareg Behailu, Tsegaye Alemayehu, Mulugeta Mengistu, Demissie Assegu Fenta.

**Writing – review & editing:** Yeshareg Behailu, Siraj Hussen, Mulugeta Mengistu, Demissie Assegu Fenta.

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
