## [Decision Letter · Decision Letter 0]

27 Jan 2022

PONE-D-22-01816Prevalence, antimicrobial susceptibility patterns and associated factors of Campylobacter infection among under-five children with diarrhea at Governmental Hospitals in Hawassa city, Sidama, Ethiopia. A cross-sectional study.PLOS ONE

Dear Dr. Fenta,

Thank you for submitting your manuscript to PLOS ONE. After careful consideration, we feel that it has merit but does not fully meet PLOS ONE’s publication criteria as it currently stands. Therefore, we invite you to submit a revised version of the manuscript that addresses the points raised during the review process.

ACADEMIC EDITOR: Please revise the manuscript according to the reviewer comments, a major revision is required.The manuscript should be revised for English editing and grammar mistakes.

We look forward to receiving your revised manuscript.

Kind regards,

Abdelazeem Mohamed Algammal, Prof, Ph.D

Academic Editor

PLOS ONE

Journal Requirements:

"The authors have not declared a specific grant for this research from any funding agency in the public, commercial or not-for-profit sectors.

The funders had no role in study design, data collection and analysis, decision to publish, or preparation of the manuscrip"

Reviewers' comments:

Reviewer's Responses to Questions

**Comments to the Author**

1. Is the manuscript technically sound, and do the data support the conclusions?

Reviewer #1: Partly

Reviewer #2: Yes

2. Has the statistical analysis been performed appropriately and rigorously? 

Reviewer #1: Yes

Reviewer #2: Yes

3. Have the authors made all data underlying the findings in their manuscript fully available?

Reviewer #1: No

Reviewer #2: Yes

4. Is the manuscript presented in an intelligible fashion and written in standard English?

Reviewer #1: No

Reviewer #2: No

5. Review Comments to the Author

Reviewer #1: Comments to authors:

-The current study is interesting; however, the authors should address the following comments to improve the quality of the manuscript:

Title:

I think the work would benefit from the title that contains the main conclusion of the study (should be derived from the conclusion). Please modify the title.

Abstract:

- The abstract must illustrate the used methods and the most prevalent results (give more hints about methods and results). Besides, rephrase the aim of the work and the main conclusion of your findings.

Introduction: (it needs to be more informative)

-Give a hint about the virulence factors, different infections caused by Campylobacter, and the mechanism of disease occurrence.

- The authors should illustrate the public health importance concerning the emergence of multidrug-resistant (MDR) bacterial pathogens that reflect the necessity of new potent and safe antimicrobial agents. Several studies proved the widespread MDR- bacterial pathogens;

Authors could add the following paragraph:

Multidrug resistance has been increased all over the world that is considered a public health threat. Several recent investigations reported the emergence of multidrug-resistant bacterial pathogens from different origins including humans, birds, cattle, and fish that increase the need for routine application of the antimicrobial susceptibility testing to detect the antibiotic of choice as well as the screening of the emerging MDR strains. You should cite the following valuable studies:

1.PMID: 33177849

2.PMID: 33188216

3.PMID: 30150182

4.PMID: 33947875

5.PMID: 32994450

6. PMID: 32497922

7.PMID: 33061472

8.PMID: 34445951

9.https://doi.org/10.1016/j.aquaculture.2021.737643

10.https://doi.org/10.1016/j.foodcont.2021.108066

-Rephrase the aim of the work to be clear and better sound.

Material and methods:

-Add this subtitle: Bacterial Isolation and identification:

•Discuss in detail the methods of isolation and identification of Campylobacter. Besides, specific references should be added.

•Add the company, city, and country of the used bacterial media and reagents that were used in the biochemical identification of isolates. Also, enumerate all used biochemical reactions.

- Antimicrobial susceptibility testing:

•Add the names of the antimicrobial classes of the tested antibiotics.

•The authors are advised to classify the tested isolates to MDR , XDR, and PDR as described by Magiorakos et al.

Magiorakos AP, Srinivasan A, Carey RB, Carmeli Y, Falagas ME, Giske CG, et al. Multidrug-resistant, extensively drug-resistant and pandrug-resistant bacteria: An international expert proposal for interim standard definitions for acquired resistance. Clin Microbiol Infect. 2012; 18:268–81. doi:10.1111/j.1469-0691.2011.03570.x.

- PCR-based dection of the antimicrobial resistance genes should be performed. Afterward, the correlation between phenotypic and genotypic multidrug resistance should be performed.

-Statistical analyses:

-Add more details about the software used in the statistical analyses.

-Results:

-Add this subtitle: Phenotypic characteristics of the recovered isolates:

•Illustrate in detail the phenotypic characteristics of the recovered Campylobacter isolates.

-Antimicrobial susceptibility testing:

•-Illustrate in a new table the occurrence of MDR (Multidrug resistance) among the recovered isolates as the following (illustrate the names of the antimicrobial classes and different antibiotics):

No. of strains%Type of resistance

R, MDR, and XDRPhenotypic multidrug resistance

(Antimicrobial classes and different antibiotics).The antibiotic -resistance genes

-The correlation (Correlation coefficient) between phenotypic and genotypic multidrug resistance should be performed.

-You should support your results with illustrating figures.

-Discussion:

- The authors are advised to illustrate the real impact of their findings.

-Illustrate the different mechanisms of antimicrobial resistance in Campylobacter.

-Conclusion

- Should be rephrased to be sounded. A real conclusion should focus on the question or claim you articulated in your study, which resolution has been the main objective of your paper?

Reviewer #2: Comments to authors:

- The current study has a significant impact, but it needs a major revision:

- The manuscript should be revised for grammar mistakes.

- Please write the scientific names of bacterial pathogens and genes in the correct form all over the manuscript and in the References section (should be italic).

-The title is broad, please modify the title.

- Add more details about the used methods and most prevalent results in the abstract.

-In the introduction: discuss the public health importance of Campylobacter spp. and their virulence determinants.

-Improve the aim of work.

Methods:

-Explain the methods of isolation and identification in detail??

-Specific references should be added to all the used methods and techniques.

- Antimicrobial susceptibility testing: Add the manufacturing company, city, and country for the used reagents and antimicrobial discs.

-PCR based detection of virulence genes and antimicrobial resistance genes in the most prevalent retrieved bacterial species should be carried out if applicable (or addresses this point in the study limitations)

-Results:

- Discuss in detail the phenotypic characters of the recovered isolates.

-PCR based detection of virulence genes and antimicrobial resistance genes in the retrieved isolates should be carried out if applicable (or addresses this point in the study limitations)

-The correlation between the phenotypic and genotypic MDR should be performed.

-Discussion:

- Please improve.

-Please improve the main conclusion of the manuscript.

6. PLOS authors have the option to publish the peer review history of their article (what does this mean?). If published, this will include your full peer review and any attached files.

Reviewer #1: No

Reviewer #2: No

---

## [Author Response · Author response to Decision Letter 0]

30 Mar 2022

CC: abdelazeem.algammal@vet.suez.edu.eg, abdelazeem.algammal@gmail.com

PONE-D-22-01816

Prevalence, antimicrobial susceptibility patterns and associated factors of Campylobacter infection among under-five children with diarrhea at Governmental Hospitals in Hawassa city, Sidama, Ethiopia. A cross-sectional study.

PLOS ONE

Dear Dr. Fenta,

Thank you for submitting your manuscript to PLOS ONE. After careful consideration, we feel that it has merit but does not fully meet PLOS ONE’s publication criteria as it currently stands. Therefore, we invite you to submit a revised version of the manuscript that addresses the points raised during the review process.

ACADEMIC EDITOR: Please revise the manuscript according to the reviewer comments, a major revision is required. The manuscript should be revised for English editing and grammar mistakes.

We look forward to receiving your revised manuscript.

Kind regards,

Abdelazeem Mohamed Algammal, Prof, Ph.D

Academic Editor

PLOS ONE

Cover letter

DEAR: JOURNAL PLOSE ONE EDITORIAL OFFICE.

We are very grateful to editors who have critically revised the manuscript for the first time and provided and raised a number of points that we believe would improve the manuscript and may allow us with points to improve its quality. Therefore, we have incorporated each and every comment of the Editors and reviewers in the revised manuscript with highlighting with yellow color. Moreover, point-by-point responses are given below to the raised queries for editors

We hope that each and every point is now addressed to the satisfaction of the Editors and reviewers

Kind Regards

Title: Prevalence, antimicrobial susceptibility patterns and associated factors of Campylobacter infection among under-five children with diarrhea at Governmental Hospitals in Hawassa city, Sidama, Ethiopia. A cross-sectional study.

PONE-D-22-01816

Journal Requirements:

Response: The comment is well taken and agrees with the Editor’s comment, more ever we tried to read and follow the PLOSE ONE style template as much as possible.

"The authors have not declared a specific grant for this research from any funding agency in the public, commercial or not-for-profit sectors.

The funders had no role in study design, data collection, and analysis, decision to publish, or preparation of the manuscript"

Response: The comment is also well taken again and we are sure that no kind of fund would be secured from any funding organization and other partners, except material support and perdiaum for data collectors was supported by Hawassa University College of Medicine and health science. This was explained in the supporting information uploaded during submission. Therefore: “The funders had no role in study design, data collection, and analysis, decision to publish, or preparation of the manuscript.”

b) State what role the funders took in the study. If the funders had no role in your study, please state: “The funders had no role in study design, data collection, and analysis, decision to publish, or preparation of the manuscript.”

Response: The comment is well taken again and we are sure that no kind of fund would be secured from any funding organization and other partners, except support by Hawassa University College of Medicine and health science for material support and perdiaum for data collectors. This was explained in the cover letter uploaded during submission. Therefore; “Any of the authors were not received a salary from any of the funders”

Response: The comment is well taken again and we are sure that no kind of fund would be secured from any funding organization and other partners, except support by Hawassa University College of Medicine and health science for material support and perdiaum for data collectors. This was explained in the cover letter uploaded during submission.

Cover letter

DEAR: JOURNAL PLOSE ONE REVIEWERS.

We are very grateful to editors who have critically revised the manuscript for the first time and provided and raised a number of points that we believe would improve the manuscript and may allow us with points to improve its quality. Therefore, we have incorporated each and every comment of the Reviewers in the revised manuscript with highlighting with yellow color. Moreover, point-by-point responses are given below to the raised queries for Reviewers.

We hope that each and every point is now addressed to the satisfaction of the Reviewers

Kind Regards

Title: Prevalence, antimicrobial susceptibility patterns and associated factors of Campylobacter infection among under-five children with diarrhea at Governmental Hospitals in Hawassa city, Sidama, Ethiopia. A cross-sectional study.

PONE-D-22-01816

Reviewers' comments:

Reviewer's Responses to Questions

Comments to the Author

1. Is the manuscript technically sound, and do the data support the conclusions?

Reviewer #1: Partly

Response: We agree with the reviewer’s comment and we tried to correct based on the comment

Reviewer #2: Yes

Response: The comment is well taken and we thank you for your encouraging comment. 

2. Has the statistical analysis been performed appropriately and rigorously?

Reviewer #1: Yes

Response: The comment is well taken and we thank you for your encouraging comment.

Reviewer #2: Yes

Response: The comment is well taken and we thank you for your encouraging comment.

3. Have the authors made all data underlying the findings in their manuscript fully available?

Response: The comment is well taken and all data underlying the findings are in our manuscript. Finally we included it in data availability section of the availability.

Reviewer #1: No

Response: The comment is well taken and all data underlying the findings are in our manuscript. Finally, we included it in the data availability in the additional information sheet

Reviewer #2: Yes

Response: The comment is well taken and we thank you for your encouraging comment.

4. Is the manuscript presented in an intelligible fashion and written in standard English?

Reviewer #1: No

Response: We agree with the reviewer comment and concern and tried to revise the language and punctuation throughout the manuscript according to the given comment

Reviewer #2: No

Response: We agree with the reviewer comment and concern and tried to revise the language and punctuation throughout the manuscript according to the given comment

5. Review Comments to the Author

Reviewer #1: Comments to authors:

-The current study is interesting; however, the authors should address the following comments to improve the quality of the manuscript

Title:

I think the work would benefit from the title that contains the main conclusion of the study (should be derived from the conclusion). Please modify the title.

Response: We agree with the reviewer’s comment and tried to modify it according to the comment given and also tried to link with a conclusion. 

Abstract:

- The abstract must illustrate the used methods and the most prevalent results (give more hints about methods and results).

Response: We agree with the reviewer’s comment and concern and tried to revise the abstract section to mention the methods and the most prevalent results (line #23-55)

 Besides, rephrase the aim of the work and the main conclusion of your findings.

Response: We agree with the reviewer’s comment and concern and tried to rephrase the abstract section to link with the conclusion part.

Introduction: (it needs to be more informative)

Response: We agree with the reviewer’s comment and tried to modify it according to the comment given.

-Give a hint about the virulence factors, different infections caused by Campylobacter, and the mechanism of disease occurrence.

Response: We agree with the reviewer’s comment and tried to modify it according to the comment given from Line # 92-110 in the manuscript

- The authors should illustrate the public health importance concerning the emergence of multidrug-resistant (MDR) bacterial pathogens that reflect the necessity of new potent and safe antimicrobial agents. Several studies proved the widespread MDR- bacterial pathogens;

Authors could add the following paragraph:

Multidrug resistance has been increased all over the world that is considered a public health threat. Several recent investigations reported the emergence of multidrug-resistant bacterial pathogens from different origins including humans, birds, cattle, and fish that increase the need for routine application of the antimicrobial susceptibility testing to detect the antibiotic of choice as well as the screening of the emerging MDR strains. You should cite the following valuable studies:

1.PMID: 33177849

2.PMID: 33188216

3.PMID: 30150182

4.PMID: 33947875

5.PMID: 32994450

6. PMID: 32497922

7.PMID: 33061472

8.PMID: 34445951

9.https://doi.org/10.1016/j.aquaculture.2021.737643

10.https://doi.org/10.1016/j.foodcont.2021.108066

Response: We agree with the reviewer’s comment and tried to address each issue as given in the comment in the main manuscript from line (#107-126), however, most of the references are not specifically related to our specific topic

-Rephrase the aim of the work to be clear and better sound.

Response: We agree with the reviewer’s comment and tried to rephrase the manuscript on the aim of the study based on the comment and to the best of our manuscript quality Line (#135-156).

Material and methods:

-Add this subtitle: Bacterial Isolation and identification:

Response: We agree with the reviewer’s comment and tried to subtitle with the given comment in the methods section line (#221) 

•Discuss in detail the methods of isolation and identification of Campylobacter. Besides, specific references should be added.

Response: We agree with the reviewer’s comment and tried to discuss the methods part in detail on the isolation and identification of Campylobacter with specific references in the methods section from lines # 221-246 based on the comment and to the best of our manuscript quality.

•Add the company, city, and country of the used bacterial media and reagents that were used in the biochemical identification of isolates. Also, enumerate all used biochemical reactions.

Response: We agree with the reviewer’s comment and tried to address the country of the media and biochemical tests used in the current study (line # 139-142). 

-Antimicrobial susceptibility testing:

•Add the names of the antimicrobial classes of the tested antibiotics.

Response: We agree with the reviewer’s comment and tried to address the country of the media and biochemical tests used in the current study (line # 139-142). 

•The authors are advised to classify the tested isolates to MDR , XDR, and PDR as described by Magiorakos et al.

Magiorakos AP, Srinivasan A, Carey RB, Carmeli Y, Falagas ME, Giske CG, et al. Multidrug-resistant, extensively drug-resistant and pandrug-resistant bacteria: An international expert proposal for interim standard definitions for acquired resistance. Clin Microbiol Infect. 2012; 18:268–81. doi:10.1111/j.1469-0691.2011.03570.x.

- PCR-based detection of the antimicrobial resistance genes should be performed. Afterward, the correlation between phenotypic and genotypic multidrug resistance should be performed.

Response: We agree with the reviewer’s comment and for the future we have plan to conduct a large scale survey using PCR-based detection the correlation between phenotypic and genotypic multidrug resistance will be performed in the future. In this study, we tried to mention it as one of the limitations of the study in the discussion section.

-Statistical analyses:-Add more details about the software used in the statistical analyses.

Response: We agree with the reviewer’s comment and added a few details about the SPSS software that we use for analysis if we tried to understand the reviewer’s raised issue in the data processing and analysis section of the manuscript line (#293-295).

-Results:

-Add this subtitle: Phenotypic characteristics of the recovered isolates:

•Illustrate in detail the phenotypic characteristics of the recovered Campylobacter isolates.

Response: We agree with the reviewer’s comment and added details of the phenotypic identification techniques that we used in this study. However; the details of the methods of isolation is also indicated in the methods part of the manuscript line (#335-351).

-Antimicrobial susceptibility testing:

•-Illustrate in a new table the occurrence of MDR (Multidrug Resistance) among the recovered isolates as the following (illustrate the names of the antimicrobial classes and different antibiotics): No. of strains%. Type of resistance R, MDR, and XDR Phenotypic multidrug resistance (Antimicrobial classes and different antibiotics).The antibiotic-resistance genes

Response: We agree with the reviewer’s comment and tried to address the classes of antibiotics used in this study in Table 5. And R, MDR, and XDR Phenotypic multidrug resistance (Antimicrobial classes and different antibiotics) in Table 6

The antibiotic-resistance genes

-The correlation (Correlation coefficient) between phenotypic and genotypic multidrug resistance should be performed.-You should support your results with illustrating figures.

Response: We agree with the reviewer’s comment and for the future we have plan to conduct a large-scale survey using PCR-based detection the correlation between phenotypic and genotypic multidrug resistance will be performed in the future. In this study, we tried to mention it as one of the limitations of the study in the discussion section.

-Discussion:

- The authors are advised to illustrate the real impact of their findings.

-Illustrate the different mechanisms of antimicrobial resistance in Campylobacter.

Response: We agree with the reviewer’s comment and we tried to rephrase and show the real impact of the findings in some parts of the discussion.

-Conclusion

- Should be rephrased to be sounded. A real conclusion should focus on the question or claim you articulated in your study, which resolution has been the main objective of your paper?

Response: We agree with the reviewer’s comment and we tried to rephrase the conclusion based on the comment given to link with the finding.

Reviewer #2: Comments to authors:

- The current study has a significant impact, but it needs a major revision:

- The manuscript should be revised for grammar mistakes.

Response: We agree with the reviewer’s comment and we tried to rephrase copy edit, topographical, and grammar mistakes throughout the manuscript based on the comment and to the best of our manuscript quality.

- Please write the scientific names of bacterial pathogens and genes in the correct form all over the manuscript and in the References section (should be italic).

Response: We agree with the reviewer’s comment and concern and tried to revise the abstract section to link with the conclusion part.

-The title is broad, please modify the title.

Response: We agree with the reviewer’s comment and concern and tried to revise the abstract section to link with the conclusion part.

- Add more details about the used methods and most prevalent results in the abstract.

Response: We agree with the reviewer’s comment and concern and tried to revise the abstract section to link with the conclusion part.

-In the introduction: discuss the public health importance of Campylobacter spp. and their virulence determinants.

Response: We agree with the reviewer’s comment and concern and tried to revise the abstract section to link with the conclusion part.

-Improve the aim of work.

Response: We agree with the reviewer’s comment and tried to rephrase the manuscript on the aim of the study based on the comment and to the best of our manuscript quality Line (#135-156).

Methods:

-Explain the methods of isolation and identification in detail??

-Specific references should be added to all the used methods and techniques.

 Response: We agree with the reviewer’s comment and tried to discuss the methods part in detail on the isolation and identification of Campylobacter with specific references in the methods section from line # 246-276 based on the comment and to the best of our manuscript quality

- Antimicrobial susceptibility testing:

 Add the manufacturing company, city, and country for the used reagents and antimicrobial discs.

Response: We agree with the reviewer’s comment and tried to address the country of the media and biochemical tests used in the current study (line # 139-142). 

-PCR based detection of virulence genes and antimicrobial resistance genes in the most prevalent retrieved bacterial species should be carried out if applicable (or addresses this point in the study limitations)

Response: We agree with the reviewer’s comment and for the future, we have a plan to conduct a large-scale survey using PCR-based detection the correlation between phenotypic and genotypic multidrug resistance will be performed in the future. In this study, we tried to mention it as one of the limitations of the study.

-Results:

- Discuss in detail the phenotypic characters of the recovered isolates.

Response: We agree with the reviewer’s comment and added details of the phenotypic identification techniques that we used in this study. However; the details of the methods of isolation is also indicated in the methods part of the manuscript line (#335-351).

-PCR based detection of virulence genes and antimicrobial resistance genes in the retrieved isolates should be carried out if applicable (or addresses this point in the study limitations)

-The correlation between the phenotypic and genotypic MDR should be performed.

Response: We agree with the reviewer’s comment and for the future, we have a plan to conduct a large scale survey using PCR-based detection the correlation between phenotypic and genotypic multidrug resistance will be performed in the future. In this study, we tried to mention it as one of the limitations of the study in the discussion section.

-Discussion:

- Please improve.

Response: We agree with the reviewer’s comment and we tried to rephrase and show the real impact of the findings in some parts of the discussion.

-Please improve the main conclusion of the manuscript.

Response: We agree with the reviewer’s comment and we tried to rephrase the conclusion based on the comment given to link with the finding.

6. PLOS authors have the option to publish the peer review history of their article (what does this mean?). If published, this will include your full peer review and any attached files.

Do you want your identity to be public for this peer review? For information about this choice, including consent withdrawal, please see our Privacy Policy.

Reviewer #1: No

Reviewer #2: No

• 

• 

• 

• 

---

## [Editor Report · Decision Letter 1]

31 Mar 2022

Prevalence, determinants, and antimicrobial susceptibility patterns of Campylobacter infection among under-five children with diarrhea at Governmental Hospitals in Hawassa city, Sidama, Ethiopia. A cross-sectional study.

PONE-D-22-01816R1

Dear Dr. Fenta,

We’re pleased to inform you that your manuscript has been judged scientifically suitable for publication and will be formally accepted for publication once it meets all outstanding technical requirements.

Kind regards,

Abdelazeem Mohamed Algammal, Prof, Ph.D

Academic Editor

PLOS ONE

Additional Editor Comments (optional):

The authors have carried out significant changes to the manuscript. They have addressed most of the suggested corrections and comments.The manuscript could be accepted.
---

## [Editor Report · Acceptance letter]

13 Apr 2022

PONE-D-22-01816R1 

*Prevalence, determinants, and antimicrobial susceptibility patterns of Campylobacter infection among under-five children with diarrhea at Governmental Hospitals in Hawassa city, Sidama, Ethiopia. A cross-sectional study*

Dear Dr. Fenta:

I'm pleased to inform you that your manuscript has been deemed suitable for publication in PLOS ONE. Congratulations! Your manuscript is now with our production department. 

Kind regards, 

on behalf of

Professor Abdelazeem Mohamed Algammal 

Academic Editor

PLOS ONE